# Fungal Flora in Asymptomatic Pet Guinea Pigs and Rabbits

**DOI:** 10.3390/ani12182387

**Published:** 2022-09-13

**Authors:** Lucia Kottferová, Ladislav Molnár, Eva Čonková, Peter Major, Edina Sesztáková, Andrea Szarková, Monika Slivková, Jana Kottferová

**Affiliations:** 1Clinic of Birds, Exotic and Free Living Animals, University of Veterinary Medicine and Pharmacy, Komenského 73, 04181 Košice, Slovakia; 2Department of Pharmacy and Pharmacology and Toxicology, University of Veterinary Medicine and Pharmacy, Komenského 73, 04181 Košice, Slovakia; 3Small Animal Clinic, University of Veterinary Medicine and Pharmacy, Komenského 73, 04181 Košice, Slovakia; 4Department of Public Veterinary Medicine and Animal Welfare, University of Veterinary Medicine and Pharmacy, Komenského 73, 04181 Košice, Slovakia

**Keywords:** fungal skin diseases, mycoses, molds, dermatophytes, small mammals

## Abstract

**Simple Summary:**

Small mammals hider a wide number of saprophytic fungi associated with dermatophytosis in young or immunocompromised humans. This raises the possibility of potential zoonotic transmission of dermatophytes in animals from pet shops. Therefore, it is recommended that routine fungal diagnostic testing in pet guinea pigs (*Cavia porcellus*) and rabbits (*Oryctolagus cuniculus domesticus*) be carried out, to detect potential zoonotic fungi. The almost complete lack of cutaneous lesions in many cases associated with the presence of such a dermatophyte on healthy mammals may increase the risk of zoonotic transmission. This study identifies the most common fungal species that occur on the skin in guinea pigs and rabbits and determines the rate of asymptomatic carriers in healthy pet animals.

**Abstract:**

Fungal skin diseases are well-recognized diseases with public health implications. The study provides a comprehensive overview and aims to determine the rate of positive fungal cultures to identify the most common fungal species in guinea pigs and rabbits and to determine the rate of asymptomatic carriers in healthy pet animals. This knowledge is essential for understanding disease transmission dynamics and epidemiological situation problems. A total of 167 animals (64 rabbits and 103 guinea pigs) were investigated in this study. The fungi of the genus *Penicillium, Rhizopus*, Mucor, Cladosporium, and *Aspergillus* were the most common in the examined animals, and they were isolated from 162 (97%) of the animals enrolled. No fungal growth was observed in 5 animals. In 15 cases (8.98%), we found pathogenic zoonotic dermatophytes (*Trichophyton mentagrophytes*), which caused several health problems in two humans in contact with affected animals. This study presents the prevalence of fungal flora in pet guinea pigs and rabbits in Slovakia.

## 1. Introduction

The dermatophytes are a group of fungi that are adapted to digest keratinous debris. They are normally located in the epidermal *stratum corneum*, hair shaft, or claw when pathogenic. The species that are adapted to animal hosts are termed “zoophilic”, but these occasionally spread to in-contact humans or other animal species. The soil-adapted geophilic dermatophytes sometimes affect animals where there is outdoor husbandry or activity [1,2]. Cutaneous mycosis, otherwise known as dermatomycosis, is caused by keratinophilic fungi. Although the group of keratinophilic fungi is large, a special subgroup known as dermatophytes can cause serious cutaneous conditions, commonly referred to as ringworm infections, in animals and humans [3]. Cutaneous mycoses are highly contagious and pose a significant global public health threat [4].

A variety of dermatophytes that live in the hair, skin, nails, and environment can infect small exotic mammals. There are significant differences between the species of animal hosts and also regional differences in the dermatophytes encountered and the disease’s prevalence.

A few studies have shown the occurrence of anthropophilic dermatophytes in the animals examined. These dermatophytes may cause health risks to humans who come into contact with animals [5,6].

*Microsporum* spp. and *Trichophyton* spp. account for most of the animal diseases, and the human-adapted anthropophilic species, such as *Epidermophyton floccosum* and *T. tonsurans,* only rarely transfer from humans to animals [1,2]. The most important dermatophytes for small mammals are *Trichophyton erinacei*, *T. mentagrophytes*, *T. quinckeanum*, *Arthroderma benhamiae*, *A. vanbreuseghemii*, and *Microsporum canis* [7]. Major changes in nomenclature preference have occurred in the last few years in the *T. mentagrophytes* complex, which includes both anthropophilic and zoophilic species. *Arthroderma benhamiae*, *A. simii*, and *A. vanbreuseghemii* are three important species that are part of the *T. mentagrophytes* complex [8,9].

Most veterinary studies deal with the prevalence of dermatophytes in the population of “healthy and sick” animals, or the prevalence of dermatophytosis among animals with a skin lesion [10,11]. Significant epidemiological changes in the prevalence and spectrum of zoophilic dermatophytes in large geographical areas typically occur over time and can only be detected if long-term epidemiological data is available. These changes can be easily overlooked in animals where infections are normally asymptomatic [12].

The general presentation of ringworm in animals is a regular and circular alopecia, with an erythematous margin and thin desquamation. Pruritus is generally absent, although it is described in some surveys in a noticeable proportion of animals. Lesions can be single or multiple and are localized to any animal part, although the anterior part of the body and the head seems part more frequently involved. Usually, there is a centrifugal spread of lesions. Multiple lesions may coalesce, while spontaneous healing at the center with regrowth of hairs is generally observed. Clinical signs of dermatophytosis in small mammals vary depending on the disease’s species, affected tissue, location, and duration. Typical macroscopic lesions include yellowish-white dry crust formation and alopecia areas around the eyes, face, and nasal bridge. Later, the disease can become generalized. In a recent dermatophytosis study involving 101 guinea pigs, lesions were distributed on the head (72%), periocular region (32%), nasal bridge (32%), ears (22%), and back (22%). In cases of dermatophytosis-infected rabbits, the most common clinical signs were erythema, scaling, alopecia, and crust formation [13,14,15]. Asymptomatic pet guinea pigs and rabbits often represent a source of fungal infections in people, even if the contact time is short [15]. Pet shops can play an important role in preventing the transmission of zoonotic dermatophytosis. Specific preventive measures such as routine screening examinations of rabbits and guinea pigs are recommended next to regular hygiene when handling animals [7]. Young rabbits and guinea pigs are often purchased in pet shops as pets for children. If these animals have zoonotic skin infections, they can spread the illness to new owners [16].

Dermatophytosis in humans is often not reported. People are usually infected by direct contact with diseased animals, or indirectly through contaminated bedding, soil, or fomites [17]. Younger or immunocompromised people with intensive contact with pets are most susceptible to infection [7]. Zoophilic fungi invading skin and hairs lead to the formation of erythema, excoriations, crusts, hyperkeratosis, alopecia, and often secondary bacterial dermatitis in humans. It is a very important fact to know that there are species differences in the zoonotic risk of animals with or without clinical signs of dermatophytosis. Transmission from human to human seldom occurs. [17,18,19,20,21].

However, over the last decades, there has been an increase in the number of fungal diseases in animals caused by opportunistic and pathogenic fungi. Opportunistic fungi have a preferred habitat independent from the living host and cause infection after accidental penetration of intact skin barriers, or when immunologic defects or other debilitating conditions exist in the host. Infection almost invariably is established only when the normal balance between the animal and the agent is disturbed [21]. Given our findings, we thought it was critical to detect fungi on the skin and hair of pet guinea pigs and rabbits, as well as identify the most common pathogens.

## 2. Materials and Methods

### 2.1. Animals

One hundred and sixty-seven animals (64 rabbits and 103 guinea pigs) were investigated in this study. Samples were taken in pet shops in Slovakia and at the Clinic of birds, exotics, and free-living animals in Košice, Slovakia from patients purchased in pet shops. The examined samples came from animals aged 4 months to 1 year. None of the patients showed clinical signs suggestive of ongoing skin disease. All the samples examined were divided into groups based on the animal species and gender, as indicated in Table 1.

### 2.2. Methods

The Mackenzie technique [22] was used to collect hair and skin scales from all animals. For one minute, the entire animal’s body was brushed. The samples were taken to the laboratory for analysis fungal cultures tests. They were incubated on the same day. Sabouraud’s dextrose agar plates containing cycloheximide 0.02%, depomycin 0.3%, and DTM medium have been used. The plates were covered with plastic to prevent dehydration, incubated in the dark at 25 °C for up to three weeks, and examined for fungal growth. Laboratory identification of the fungal isolates was based on macroscopic and microscopic characteristics. Macroscopic features evaluated included the color colonies, the texture of the colonies, whether the colonies were fluffy, powdery, cottony, velvety, etc., whether the hyphae were radiating at the margins, and whether the colonies were folded/grooved, or furrowed. A small portion of the test colony was picked with a sterile needle and placed on a drop of lactophenol cotton blue on a clean microscope slide to examine the isolates for microscopic features. Then the slide was covered with a coverslip and viewed under the microscope (using ×40 and ×100 objectives) for the presence, shape, arrangement, and relative abundance of micro and macroconidia. The microscopic characteristics of the hyphae were also noted.

## 3. Results

The prospective cross-sectional study consisted of 167 asymptomatic pet guinea pigs and rabbits. From all the examined samples, we cultivated fungi of 12 genera. The occurrence of fungi isolated from examined animals is presented in Table 2. In 15 cases, we found the zoonotic dermatophyte *Trichophyton mentagrophytes*. The most frequently detected were environmental molds of *Penicillium*, *Rhizopus*, *Mucor*, *Cladosporium*, and *Aspergillus fungi*. *Penicillium* spp. was isolated from 101 examined animals, *Rhizopus* spp. was found in samples obtained from 58 animals, *Mucor* spp. was detected in 54, and *Cladosporium* spp. in 40 animals. Out of all the examined animals, 39 were positive for *Aspergillus* spp. In addition to the most frequent genera, we also cultivated fungi of the genus *Alternaria, Fusarium*, *Humicola*, and *Paecilomyces.* of the 167 examined samples, only 5 were negative.

### 3.1. Clinical Case 1

Two six-month-old male guinea pigs were referred to the clinic. The guinea pigs were purchased at a pet store two months ago. An eleven years old girl developed numerous circular lesions of varying sizes on her chest (Figure 1). Analysis of skin scratches from the child’s affected area confirmed the suspicion of zoonotic dermatophytosis. The guinea pigs’ clinical examination revealed no signs of dermatological problems; the skin was intact, and no lesions were visible (Figure 2). There were no additional health issues mentioned in the animal patient’s history. We collected samples from a guinea pig’s entire body as well as hairs and sent them to be cultured and examined. *Trichophyton mentagrophytes* was found in both guinea pigs, according to the results. We have chosen Imaverol^®^ (*enilconazole* 100 mg/mL) for the treatment of guinea pigs in the form of four baths, three days apart. We repeated the fungal culture examination of the collected samples after the therapy was completed. After 12 days, the results revealed a negative finding in animals.

### 3.2. Clinical Case 2

A 25-year-old man with severe obesity had suffered from tinea corporis with a round lesion on his belly for several weeks (Figure 3). He had contact with a guinea pig at home. The male owner, examined by a dermatologist, was positive for *Trichophyton mentagrophytes*, which was subsequently confirmed also in our guinea pig patient. Twelve months old guinea pig was purchased from the pet shop. The clinical examination of the guinea pig showed no signs of any dermatological problems; the skin was intact, and no lesions were visible. The animal’s medical history revealed no additional health issues. Samples were obtained from the entire body for fungal cultures testing, and hairs were also analyzed. This clinical case points to zoonotic transmission of *Trichophyton mentagrophytes* between the owner and his animal.

In this study, Imaverol^®^ (*enilconazole* 100 mg/mL) was utilized in the form of four baths given three days apart for the therapy of the guinea pig. A control examination was performed after the treatment. Cultivation no longer confirms the existence of pathogens.

## 4. Discussion

Fungi are extremely resistant to the effects of the external environment, and their individual species are abundant in nature in soil, surface water, air, plants, various inanimate objects, moldy food, but also on animal skin, and hair. Few of the millions of fungal species fulfill four basic conditions for infecting animals and humans: high-temperature tolerance, ability to invade the human host, lysis and absorption of human tissue, and immune system resistance. Because animals’ sophisticated immune systems evolved in constant response to fungal challenges, the invasive fungal disease is unusual in previously healthy individuals.

### 4.1. Dermatophytosis in Small Mammals

Predisposing factors for dermatophytoses in pet mammals include stress, overcrowding, immunodeficiency, poor husbandry, malnutrition, concurrent underlying diseases, and pregnancy [23]. The susceptibility of dermatophytes to infect the host depends on the dermatophyte species, the number of infection spores, virulence factors, and the immune status of the host [24]. A study by Kraemer et al. [23] appears to show no gender predisposition for dermatophyte disease in rabbits or guinea pigs. This statement also correlates with our results, where we did not record any gender or age predisposition in our examined animals. A study in Switzerland [25], evaluating guinea pigs with suspected dermatophytosis, could not demonstrate that diseased animals have an age predisposition, but in the study of Vangeel et al. [26], diseased guinea pigs and rabbits were significantly younger than those with negative cultures, and healthy animals from a control population.

Unfortunately, clinical signs are not always visible and therefore the detection of fungal skin disease is more difficult detect, a shown also in our study, where all examined animals did not show any clinical signs of the disease. Asymptomatic exotic pet animals often represent a source of fungal infections in people, even if the contact time is short [16]. The results of the study in Germany indicate that up to 27% of Dutch pet shops are selling *T. mentagrophytes-*positive rabbits and guinea pigs. None of the animals showed clinical signs and were, therefore, asymptomatic carriers [19,20], which also confirms our clinical results A total of 115 guinea pigs and 104 rabbits without skin lesions were sampled by Vangeel et al. [26]. Mycological examination confirmed the occurrence of *Trichophyton mentagrophytes var. mentagrophytes* in 4 guinea pigs and 4 rabbits. *Microsporum canis* was not isolated from any animal. In our study, we did not find any *Microsporum canis*-positive animals.

*T. mentagrophytes* is the most common dermatophyte isolated from laboratory and farmed guinea pigs and rabbits [27,28], with guinea pigs being affected more frequently than rabbits [29]. The study by Kraemer et al. [30] also showed that dermatophytosis is a common disease among pet guinea pigs, while pet rabbits are affected less frequently, which correlates with our results. Most of the data came from rabbit farms, where the percentage of animals with and without clinical signs positive for T. mentagrophytes ranges from 30 to 79.8% [31,32]. This may be due to a high rate of animal fluctuation and high infection pressure. There are several reports of guinea pigs and rabbits being asymptomatic carriers of dermatophytes. In a study evaluating healthy animals from private households, pet shops, and laboratories, 3.5% of the guinea pigs and 3.8% of the rabbits carried *T. mentagrophytes* [26]. We found the incidence of *Trichophyton mentagrophytes*-positive rabbits in 4 cases out of 64 examined. From 103 examined guinea pigs we recorded the occurrence of *Trichophyton mentagrophytes* in 11 cases. Other authors report healthy rabbits being positive not only for *T. mentagrophytes* but also for other *Trichophyton *spp. and *Microsporum *spp. [33,34,35,36]. In our study, we isolated *Trichophyton mentagrophytes* in 15 animals from 167 examined. Many cases of transmission of dermatophytes from asymptomatic carriers have been confirmed, but it should not be forgotten that if some animals may remain asymptomatic for an extended period, certainly this is not the case for all or even most. Zoophilic organisms usually present with a symptomatic inflammatory response, and they are less likely to lead to asymptomatic carriage.

To assess the risk of acquiring a zoonotic infection from animals, the study of the prevalence of dermatophytes on rabbits and guinea pigs in pet shops has been investigated in the Netherlands as well. *M. canis* (54.8%), *T. mentagrophytes* (14.7%), and *T. verrucosum* (8.1%) were the species that were found most often [37]. In a study done in Brazil, *T. rubrum* (33.3%), *T. tonsurans* (13.1%), *T. verrucosum* (11.1%), *T. interdigitale* (9.1%), and *T. mentagrophytes* (6.1%) were found to be the most common types of dermatophytes [38]. A study made by Chang [39] confirmed the prevalence of dermatophyte infection in pet rabbits at 13.9%. In the present study, the prevalence was higher than those detected in veterinary clinics, pet shops, and pet cafés reported in Southern Italy (3.29%), the Netherlands (3.80%), Chile (7.14%), and Thailand (12.1%) [38,40,41,42]. By contrast, the prevalence was lower than those detected in veterinary clinics reported in Northern Italy (27.78%) and Bangladesh (88.89%) [43,44].

The epidemic spread of *T. benhamiae* in Europe was probably the most important event in recent years in the epidemiology of zoonotic dermatomycoses [10,12]. In a study in 2016, *T. benhamiae* was isolated from more than 90% of the guinea pigs sampled in 15 pet shops in Berlin [45]. In a study in 2019, in which no guinea pigs from pet shops were sampled, only 55.4% of the animals of *T. benhamiae* were found. In both studies, 87% of patients were asymptomatic carriers. A Danish study [46] confirmed that the animals from conventional breeding are more severely affected than those from private breeding. Recent data shows that *T. benhamiae* is responsible for almost 90% of the dermatophytoses in guinea pigs [30]. The results of our work did not show an occurrence of *Trichophyton benhamiae* in Slovakia.

### 4.2. Dermatophytosis in Humans

Tinea corporis in people is more common than is reported. Although the incidence is low, sometimes tinea is misdiagnosed and underreported, and the differential diagnosis may include seborrheic dermatitis or atopic dermatitis. Tinea should be suspected in people with alopecia, pruritus, and/or persistent epithelial desquamation, and the skin lesion should be investigated from a mycological point of view. In two human clinical cases we reported above, tinea was misdiagnosed with seborrheic dermatitis, and only after culture examination was the cause of dermatitis detected.

Guinea pigs and rabbits have been identified as potential zoonotic sources of dermatophytes affecting humans, especially children and immunosuppressed individuals [23,24,25,26,27]. There are several reports of infections among pet shop employees or new owners of infected animals. In our study, we observed marked clinical signs of dermatophytosis in two guinea pig owners. We confirmed the presence of *Trichophyton mentagrophytes* in guinea pigs of these owners. Recently, a significantly increased prevalence of positive guinea pig fungal cultures was reported by Halsby et al. [16]. 11% of the owners reported dermatophytosis signs in family members. This percentage was the highest among guinea pig owners. guinea pigs are asymptomatic carriers that may pose a higher zoonotic risk than dogs, cats, and rabbits. An epidemiological survey in Germany describes 377 people with dermatophytosis. The most frequent species isolated were *M. canis* (54.8%), *T. mentagrophytes* (14.7%), and *T. verrucosum* (8.1%) [37]. In South America and Malta, 121 cases proved to be *T. rubrum*, 109 *M. canis*, 80 *T. mentagrophytes*, 27 *Microsporum gypseum,* 17 *Epidermophyton floccosum*, and 1 *M. persicolor* from 371 human cases [47]. In a Brazilian study, *T. rubrum* (33.3%), *T. tonsurans* (13.1%), *T. verrucosum* (11.1%), *T. interdigitale* (9.1%), and *T. mentagrophytes* (6.1%) were found to be the most common dermatophyte species in people [48].

### 4.3. Saprophytic Fungi

The species of *Aspergillus* and *Penicillium* anamorphs are very important, and the most common fungi found in nature. According to Kupsch et al. [45], nonpathogenic fungi such as *Aspergillus*, *Penicillium*, *Chrysosporium*, *Fusarium*, *Trichophyton*, *Microsporum*, *Acremonium*, *Mucor*, *Arthroderma*, *Malbranchea*, *Chaetomium*, *Ulocladium*, *Verticillium*, *Paecilomyces*, and S*copulariopsis* predominate in the genera. These results are consistent with our results, as we demonstrate that *Penicillium *spp. is the fungi most commonly occurring in the skin and hair of guinea pigs. In our research, *Penicillium *spp. was isolated from 101 examined animals out of 167 examined. A study by Mircean et al. [49] includes 403 rodents of different ages and sexes examined for dermatological diseases. Pathogenic dermatophytes were detected in 9 (4.5%) of 200 animals, *Epidermophyton* species in 2 (1%), and *Scopulariopsis* species in 7 (3.5%). Fehr [7] isolated saprophytic fungi from 151 (75.5%) of 200 animals. *Mucor* species were isolated in 80 (40%), *Aspergillus species* in 34 (17%), *Penicillium species* in 26 (13%), *Alternaria species* in 17 (8.5%), *Cladosporium species* in 16 (8%), *Rhizopus species* in 12 (6%), and *Chrysosporium species* in 2 (1%). No fungi growth was observed in 40 (20%) guinea pigs, while in our research results from 5 animals were negative. In the zoological park of Rome, 115 animals were analyzed for the presence of keratinophilic flora. Among the accompanying fungal flora, A*lternaria*, *Aspergillus*, *Cephalosporium*, *Cladosporium*, *Geotrichum*, *Penicillium*, and *Scopulariopsis* were the most widespread genera [50]. The study by Andrews et al. [51] found a wide range of non-pathogenic fungi, that are common in the environment where pet guinea pigs live, such as on the ground (*Mucor* and *Rhizopus* species), in the decomposing matter (*Penicillium* and *Rhizopus species*), or everywhere (*Aspergillus species*). In our study, we found fungi of 12 genera. *Penicillium*, *Rhizopus*, *Mucor*, *Cladosporium*, and *Aspergillus* fungi were the most commonly found.

## 5. Conclusions

Dermatophytosis is a disease of public health and economic importance that is reported in practically all regions of the world. An appropriate veterinary examination of every new animal is highly advised to prevent recurrent or new infections. Considering the results of our work, we also recommend a preventive mycotic examination of the skin scales and hair, as the incidence of occurrence of zoonotically transmitted pathogens has been shown in almost nine percent of our examined asymptomatic pet animals. Veterinarians should also inform pet shop employees, especially guinea pigs and rabbit handlers, about dermatophyte infection risks and make preventive fungal cultures from animals in pet shops. Control of this disease has recently gained attention due to an increase in drug resistance in many causative dermatophytes, as well as because treatment of dermatophytosis, particularly in animals, is typically costly and time-consuming. Thus, the development of an anti-dermatophytes vaccine may provide a solution to reduce these disadvantages while also limiting dermatophyte transmission between humans and animals. We also strongly recommended closer interdisciplinary cooperation between animal owners, veterinarians, dermatologists, epidemiologists, and animal shops.

## Figures and Tables

**Figure 1 animals-12-02387-f001:**
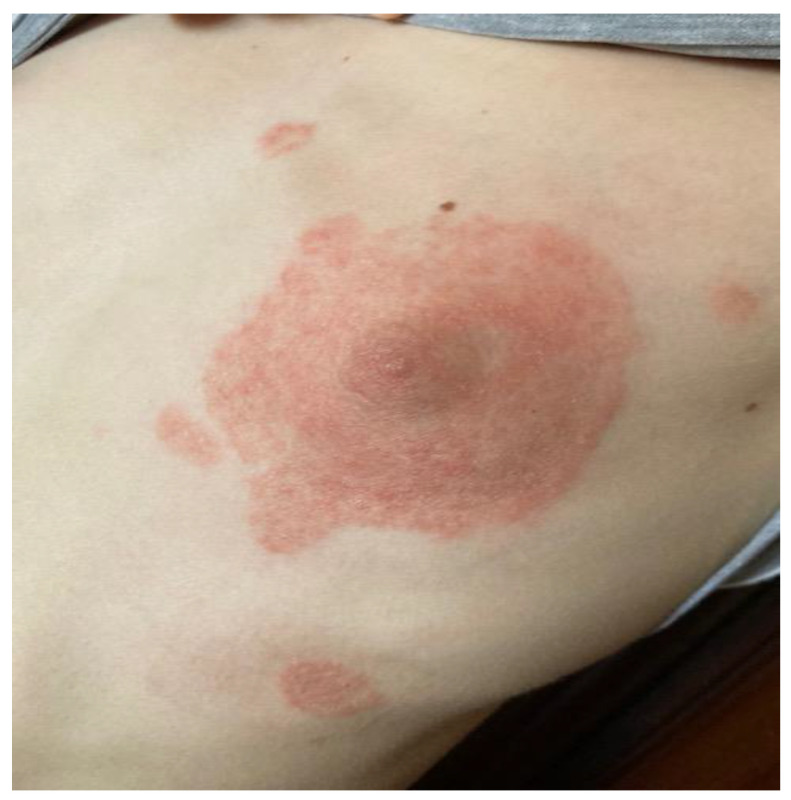
*Trichophyton mentagrophytes* caused dermatophytosis in an 11-year-old girl.

**Figure 2 animals-12-02387-f002:**
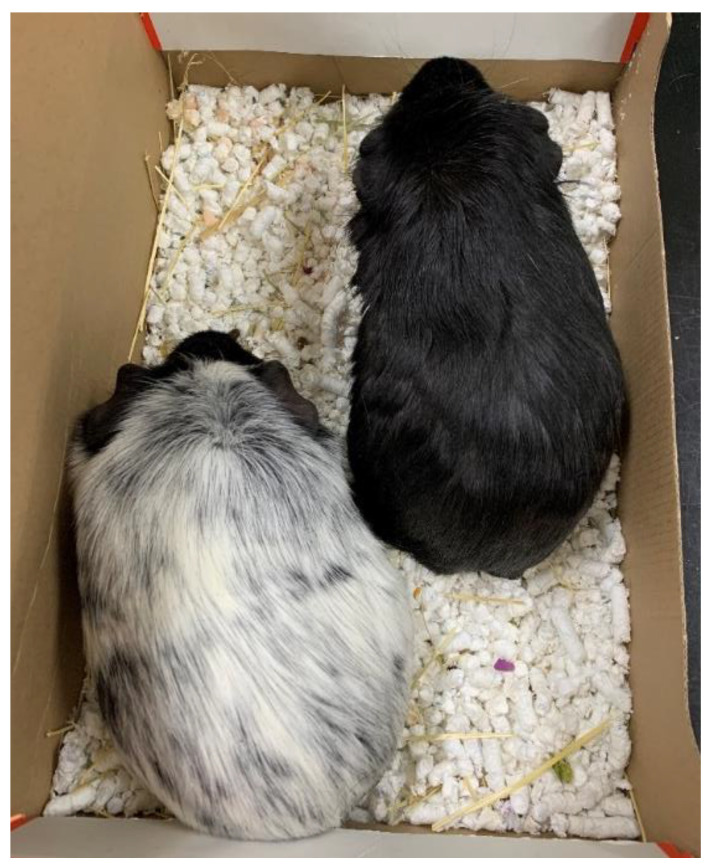
Guinea pigs affected by dermatophytosis with no clinical signs.

**Figure 3 animals-12-02387-f003:**
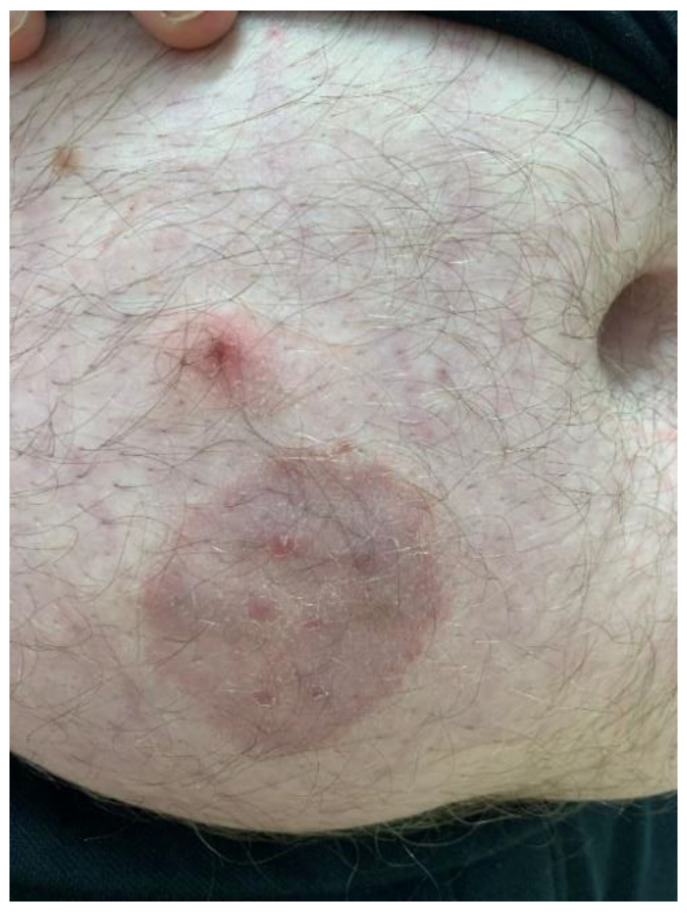
A round lesion on the belly of a male guinea pig owner.

**Table 1 animals-12-02387-t001:** Tested animals based on gender.

Rabbits		Guinea Pigs	
Gender	Number of cases/Percentage	Gender	Number of cases/Percentage
Male	33/51.6	Male	56/54.4
Female	31/48.4	Female	47/45.6

**Table 2 animals-12-02387-t002:** Genera of fungi isolated from the skin and hair of 64 rabbits and 103 guinea pigs.

	Rabbits		Guinea Pigs	
Fungi Genera	No. of Cases	Percentage	No. of Cases	Percentage
** *Trichophyton mentagrophytes* **	4	6.25	11	10.68
***Alternaria* spp.**	7	10.93	9	8.74
***Aspergillus* spp.**	18	28.12	21	0.39
***Cladosporium* spp.**	15	23.43	25	24.27
***Fusarium* spp.**	0	0	2	1.94
** *Humicola fuscoatra* **	0	0	1	0.97
***Mucor* spp.**	20	31.25	34	33.01
***Paecylomyces* spp.**	1	1.56	0	0
***Penicillium* spp.**	45	70.31	56	54.37
***Rhizopus* spp.**	25	39.06	33	32.04
**Negative**	3	4.69	2	1.94

## Data Availability

Not applicable.

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
