# Peer review of "Fungal Flora in Asymptomatic Pet Guinea Pigs and Rabbits"

_animals, 2022, doi:10.3390/ani12182387_

Round 1

Reviewer 1 Report

My suggustion: Accept after minor review. 1. Is the result of fungal identification by morphological observation accurate enough? Since the molecular method is more useful and effective. 2. In the section of dicussion, the authors only list many examples and numbers, but lack of some deep thinking, such as the reason why some fungi are more common than others, etc.

Author Response

Dear Reviewer 1,

Thank you for the thoughtful comments and advice! We have carefully addressed the reviewer´s suggestions and in doing so feel the manuscript is substantially strengthened. We have addressed the reviewer’s specific concerns as follows:

Point-by-point Response to Reviewer 1 Comments

Reviewer 2 Report

Review: SKIN FUNGAL FLORA IN ASYMPTOMATIC PET GUINEA PIGS AND RABBITS

This manuscript is interesting and brings new, original data. The results are important not only for experts, but also have a significant value for the health of the public (owners and potential owners of small mammals and workers in pet centers).

 Conclusion: major corrections, a discussion part and conclusions need rework.

 Terminology:

Line 24–25: This study identifies the most common skin fungal species in guinea pigs and rabbits

The authors mentioned skin fungal species, but most of the genera found in this study were identified by them as environmental fungi in their results: environmental mold of the molds Penicillium, Rhizopus, Mucor, Cladosporium, and Aspergillus fungi (line 133–134). The authors should clarify whether these are really skin species or just species that occur in the environment, but cannot be identified as strictly skin species (like dermatomycoses). Some species of mentioned genera are also associated with e.g. fruit/ vegetables which small mammals feed on and not with mammals skin, but i tis not mentioned in this paper. Thus, identification on species, not only genera level (table 2), can helps to distinguish them and precise terminology is really important.

 Methodology:

Line 113: histological analysis is mentioned without any details (or is it only mistake of naming of using analysis). What processing method was used (cryosectioning or standard paraffin embedding procedure, type of microtome, used staining e.g. hematoxylin-eosin). Line 147 histological examination of a swab – swab (smear) is not a histological, but a cytological method.

 Results:

Table 2: Scopulariopsis and Verticillium are not present in this table, but they are mentioned in line 138

 Gender of animals are mentioned in methodology, but not in the result part. Even if there were no significant differences between the sexes of infected with different types of fungi, a brief mention of similar numbers is appropriate.

 Line 152–153: was found in both of our patients – patients evoke people (girl) and this paper will also read doctors, and not only veterinarians. Thus, use animals or guinea pigs. Line 169 and 171 patient (guinea pig?)

 Line 153 and 173: Who has Imaverol been used for? In guinea pigs, humans or both?

 Line 197–198 It was concluded 197 that guinea pigs are asymptomatic carriers that pose a higher zoonotic risk than dogs, cats, 198 and rabbits. This is a strong claim, research should be conducted under the same conditions in the same countries. Therefore, it is better to write "may be" or "is likely".

 Line 200–206 and 210–227, 249–269: this part is only a statement, not a discussion. The discussion should be enriched by the comparison of your results with those of previous studies. E.g. Kraemer et al. [23] appears to show no gender predisposition for dermatophyte disease. Has this result also been confirmed in Slovakia? If no, why? In this way, compare all your  results. Sentences which are incomparable move to the introduction part.  

 4.3 Lines 171–298 different types of fungi are mixed here. It is necessary to revise this part and sort out what the authors write about.

 Conclusions: the authors generally advise what other professionals should do without mentioning their own specific results. We already knew this before this study, the conclusions should be based on the authors' own results.

Type errors and English: see my file, some type errors, problematic or unclear parts are highlighted in yellow.

Remember, that original address names should be used even with hooks and commas, do not anglicize local names, especially not in the authors' addressess. MDPI has no problem with characters unique to some languages ​​and it is better to be precise.

 References: double check the correct format, there are many errors (e.g. - / –, only the first few are marked by me)

Author Response

Dear Reviewer 1,

Thank you for the thoughtful comments and advice! We have carefully addressed the reviewer´s suggestions and in doing so feel the manuscript is substantially strengthened. We have addressed the reviewer’s specific concerns as follows:

Reviewer 3 Report

The Reviewer finds the work under examination to be relatively high interest to the target audience.  Several important caveats below.  

Line 51/52 reads as a non sequitur and requires rewording.  Likewise, line 54/55 presents as a circular thought.  Line 82 needs a more holistic definition of "exotic pet animals" if it is to provide useful context.  Clinical case 2 beginning at line 165 appears to describe zoonotic transmission of Trichophyton mentagrophytes from animal to human but fails to definitively confirm the point.  Line 187 appears to suggest that tinea should be suspected in an individual presenting with alopecia and / or pruritus.  This is an over-broad assertion inasmuch as the various alopecias and itch-disorders comprise etiologies of many forms, very few of which are associated with tinea.  While some animals may remain asymptomatic for an extended period, certainly this is not the case for all or even most - therefore line 231 constitutes a place where it would be appropriate to at least briefly address the typical latency period for the various dermatophytic disorders being discussed.  In the Conclusions section (beginning line 301) there appears no mention of prophylaxis of dermatophytosis.  Would this not constitute a rational approach to preventing the problem under discussion?  

Author Response

Dear Reviewer 2,

Thank you for the thoughtful comments and advice! We have carefully addressed the reviewer´s suggestions and in doing so feel the manuscript is substantially strengthened. We have addressed the reviewer’s specific concerns as follows:

Round 2

Reviewer 2 Report

The authors corrected their manuscript well. 

Author Response

Thank you very much